# Do Organochlorine Contaminants Modulate the Parasitic Infection Degree in Mediterranean Trout (*Salmo trutta*)?

**DOI:** 10.3390/ani13182961

**Published:** 2023-09-19

**Authors:** Anna Monnolo, Maria Teresa Clausi, Filomena Del Piano, Mario Santoro, Maria Lorena Fiorentino, Lorella Barca, Giovanna Fusco, Barbara Degli Uberti, Luigia Ferrante, Raffaelina Mercogliano, Maria Carmela Ferrante

**Affiliations:** 1Department of Veterinary Medicine and Animal Productions, University of Naples Federico II, Via Federico Delpino 1, 80137 Naples, Italy; anna.monnolo@unina.it (A.M.); filomena.delpiano@unina.it (F.D.P.); raffaelina.mercogliano@unina.it (R.M.); 2Experimental Zooprophylactic Institute of Southern Italy, Calabria Section, 88100 Catanzaro, Italy; mariateresa.clausi@izsmportici.it; 3Department of Integrative Marine Ecology, Stazione Zoologica Anton Dohrn, 80121 Naples, Italy; mario.santoro@szn.it; 4Environmental Research Center, Istituti Clinici Scientifici Maugeri IRCCS, 27100 Pavia, Italy; marialorena.fiorentino@icsmaugeri.it; 5Experimental Zooprophylactic Institute of Southern Italy, Calabria Section, 87100 Cosenza, Italy; lorella.barca@izsmportici.it; 6Experimental Zooprophylactic Institute of Southern Italy, 80055 Portici, Italy; giovanna.fusco@izsmportici.it (G.F.); barbara.degliuberti@izsmportici.it (B.D.U.); 7Department of Biomedical Sciences, Humanitas University, 20072 Milan, Italy; luigia.ferrante@st.hunimed.eu

**Keywords:** organochlorine contaminants, parasite infection, brown trout, risk evaluation

## Abstract

**Simple Summary:**

Living organisms may be simultaneously exposed to several stressors. Trout are a clear example of this scenario, being usually exposed to chemical contaminants and parasites. Moreover, it is also an intensely farmed fish, often caught for human consumption. Thus, trout may be considered a sentinel species, and the related evidence is a matter of importance. We looked at a sample of wild brown trout from a protected area of Southern Italy. Different numbers of gastro-intestinal helminths and concentrations of organochlorine pollutants were detected among sample units. In particular, a negative correlation between the presence of parasites and the concentration of polychlorinated biphenyls was assessed, suggesting that these organochlorine pollutants can affect parasite survival. Our results provide a contribution to the advancement of the knowledge of the interplay among environment, parasites, and host.

**Abstract:**

We investigated the occurrence of organochlorine pollutants (OCs) in the muscle of brown trout and evaluated their potential modulation of parasite infection. The toxicological risk for consumer health was assessed, too. Trout were collected from the Sila National Park (Calabria region, South of Italy). The highest concentrations emerged for the sum of the 6 non-dioxin-like (ndl) indicator polychlorinated biphenyls (Σ6ndl-PCBs), followed by the 1,1,1-trichloro-2,2-di(4-chlorophenyl)-ethane (DDT), dioxin-like PCBs, hexachlorobenzene (HCB), and dieldrin. Measured on lipid weight (LW), the mean value of Σ6ndl-PCBs amounted to 201.9 ng g^−1^, that of ΣDDTs (the sum of DDT-related compounds) to 100.2 ng g^−1^, with the major contribution of the DDT-metabolite *p,p’*-DDE which was detected in all sample units (97.6 ng g^−1^ on average). Among dioxin-like congeners, PCB 118 showed the highest mean concentration (21.96 ng g^−1^ LW) and was detected in all sample units. Regression analysis of intestinal parasites on OC concentration was performed, controlling for two potential confounding factors, namely sex and sexual stage. The results evidenced the existence of interactions between the dual stressors in the host-parasite system in the wild. A negative and statistically significant correlation was estimated, suggesting that OCs may decrease parasite infection degree. Regarding the toxicological risk evaluation, OC concentrations were consistently below the current European Maximum Residue Limits.

## 1. Introduction

Polychlorinated biphenyls (PCBs) and organochlorine pesticides (OCPs) are organochlorine compounds (OCs) widely used in the past that still persist in the environment due to their high chemical stability and lipophilicity. Aquatic organisms living in a contaminated ecosystem undergo OC biomagnification processes [1], resulting in high OC concentrations in the case of predatory species [2].

OCs are endocrine disruptors that induce a wide range of adverse effects, including alterations of reproductive, nervous, and immune systems, in both humans and wild animals, especially aquatic species [3]. In general, exposure to OCs leads to a suppression of the immune system’s capability in fish [4,5,6,7,8,9]; in some cases, it leads to stimulation [10,11] or lack of adverse effects [12]. The extent of stimulus or suppression may be influenced by the exposure level and physiological factors, including sex and age [13,14]. For instance, Duffy and coauthors observed that the immunodepression increased in young Japanese medaka (*Oryzias latipes*) exposed to PCB 126, whereas it was unaffected in mature fish [15].

Digenean trematodes are the most common eukaryotic parasites in the aquatic environment with roughly 25,000 species identified and widely distributed in European ecosystems [16,17]. In particular, the intestine of freshwater teleost commonly harbors trematodes of the genus *Crepidostomum* [18]. Some authors suggest that these parasites do not cause relevant damage to fish [19], just exploiting the intestinal fluids [20].

Chemical xenobiotics and parasites may co-contaminate living beings [21], determining a host response that may depend on their interaction [22]. Recent research based on donkey milk [23] provided evidence of a positive correlation between intestinal parasitic infection and OC contamination, the latter being probably responsible for weakened immune system defense. Henríquez-Hernández et al. [24] hypothesized that intestinal parasites in dogs might reduce OCs bioavailability and their body burden in the host by bioaccumulating OCs introduced through food intake by the host itself. OCs may impair parasite success in reaching the host [25] and decrease the population density of those parasites with complex life cycles [22] and free-life stages [26].

The main goal of the present study was to investigate the potential interaction between OCs and parasites in freshwater fish. We analyzed wild brown trout (*Salmo trutta*) from a protected area of Southern Italy for the presence of parasites and OCs contamination of the fillet. We looked at brown trout because it is a predatory species, common in freshwaters and coastal marine fish faunas, endemic in Europe [27]. Trout are frequently exposed to OCs [28,29,30] and gastro-intestinal (GI) helminths [31,32]. In rainbow trout, these parasites may induce moderate pathogenicity, such as gut inflammation and malabsorption as well as decreased hemoglobin level and hematocrit [33].

Since trout are intensely farmed and often caught for human consumption [34], their contamination by several PCBs and OCPs could be a cause of concern for human health. We thus also evaluated the ensuing risk to human health due to the consumption of contaminated fillets.

## 2. Materials and Methods

### 2.1. Study Area and Fish Collection

Fish were collected in the Trionto River, in the municipality of Longobucco (Cosenza province of Calabria region in Southern Italy) (Figure 1). The Trionto River with a length of about 40 km arises in the province of Cosenza, from the mountains of the Sila National Park, and flows east, entering the Ionian Sea. This site has long suffered from inadequate urban waste disposal (including illegal dumps or not compliant landfills located very close to riverbeds), inefficient wastewater treatment plants, and the presence of some areas of concern for environmental remediation, including a National Interest Site (Rapporti ISTISAN 16/9).

A total of 87 brown trout were collected by electrofishing with an IG600 Scubla & C. Snc equipment in the daytime in June and July 2013. Electrofishing permits (n. 2283, 4 April 2013) were issued by the Parco Nazionale della Sila on request by the Istituto Zooprofilattico Sperimentale del Mezzogiorno (n. 0001933, 13 February 2013). Immediately after being caught, fish were euthanized by gills cutting to ensure death while still unconscious according to the recommendations by the European Directive 2010/63 and Jung-Schroers et al. [35]. Specimens were transported to the laboratory under refrigerated conditions (4–8 °C) and then stored at −20 °C until the analyses were performed. Soon after thawing, biometrical data, length (cm), and weight (g) of trout were recorded. Fish were weighed using an electronic balance to the nearest 0.1 g. The length was measured considering the fork length (FL), that is, from the tip of the snout to the fork of the tail to the nearest 0.1 cm, with the help of a digital caliper. The gonads of examined fish were fixed in 10% formalin, embedded in paraffin wax, sectioned at 6 μm thick, and stained with hematoxylin and eosin. Then, slices were microscopically examined to evaluate sex and gonadal maturity stage (mature and immature trout), on the basis of criteria by Hajirezaee et al. [36]. Prevalence, abundance, and mean abundance of infection were according to Bush et al. [37].

### 2.2. Analytical Sample Preparation

We started with 87 pieces of the dorsal muscle of as many trout. However, since each piece of trout was too small to be analyzed individually, we formed 22 statistical units, each one consisting of 3–5 trout pieces obtained from trout as homogenous as possible in terms of weight, sex, sexual developmental stage, and presence or absence of GI parasites. The length of each sample unit is the average fork length of the trout corresponding to these pieces.

### 2.3. Analytical Procedure

Each sample unit was examined for the presence of 5 OCPs, 15 non-dioxin-like (ndl)-PCBs, and 8 dioxin-like (dl)-PCBs. As regards OCPs, the concentrations of *p,p’*-DDT (1,1,1-trichloro-2,2-di(4-chlorophenyl)-ethane) and its main metabolites *p,p’*-DDE (1,1-dichloro-2,2-bis(4-chlorophenyl)-ethylene) and *p,p’*-DDD (1,1-dichloro-2,2-bis-(4-chlorophenyl)-ethane) as far as HCB (hexachlorobenzene) and dieldrin were determined. In the following, we reported as ΣDDTs the sum of DDT-related compounds. As regards ndl-PCBs, we measured the six indicator congeners (IUPAC nos. 28, 52, 101, 138, 153, and 180) and the other nine non-indicator ndl-PCBs (namely, IUPAC nos. 66, 74, 128, 146, 170, 183, 187, 194, and 196). The sum of the indicator PCBs was reported as Σ6ndl-PCBs, while Σ15ndl-PCBs denotes the sum of all ndl-congeners. Σ6ndl-PCBs is indicative overall of ndl-PCBs pollution and has been adopted since 2011 by EU legislation to set tolerance limits in several foodstuffs (EU Reg. no. 1259/2011). The other non-indicator ndl-PCBs were examined because previously detected in foods of animal origin and other biotic matrices [38,39]. Finally, among the dl-PCBs we measured 3 non-ortho congeners (IUPAC nos. 77, 126, 169) and 5 mono-ortho congeners (IUPAC nos. 105, 118, 156, 157, 167), selected because of their overall high toxic equivalency factors (TEFs). Their concentrations were summed up and reported as Σ3dl-PCBs and Σ5dl-PCBs, respectively.

#### 2.3.1. Chemical Analysis

Regarding the OCs analysis, for the extraction procedure, including the separation of the analytes from the lipid fraction and the purification of the final extracts, we followed the method described by Ferrante et al. [40]. Briefly, each sample unit (roughly 3 g) was homogenized and manually cold-extracted using petroleum ether/acetone (1:1, *v*/*v*). After centrifugation, the extract was passed through a glass tube packed with anhydrous sodium sulfate. Extraction was repeated and additional extract was put into the column. The eluate was then evaporated to dryness by rotavapor, and the lipid content was gravimetrically determined. Separation of OCs from the lipid fraction was performed using a combined system of Extrelut-3/Extrelut-1 cartridges (Merck Kga A Darmstadt, Germany), previously modified by adding 0.36 g C-18 Isolute (40 to 60 mesh Merck Kga A Darmstadt, Germany). The analytes were eluted with acetonitrile and cleaned up on a glass column containing 2.5 g of heat-activated Florisil (60/100 mesh Supelco, Bellefonte, PA, USA) to separate OCs in three fractions. The column was eluted with 30 mL of n-hexane, 25 mL of n-hexane/toluene (80:20, *v*/*v*), and 30 mL of n-hexane/toluene/ethyl acetate (180:19:1, *v*/*v*/*v*) in subsequent order. The first fraction contained all PCBs, HCB, and *p,p’*-DDE, the second one contained *p,p’*-DDT and *p,p’*-DDD, while dieldrin was obtained with the third fraction. The three fractions were then concentrated to a small volume and PCB 209 was added as an internal standard (IS) for instrumental analysis. The choice of using PCB 209 as IS was due to its almost absence in the environment [41]; a more detailed explanation has been previously reported [38].

#### 2.3.2. Instrumental Analysis

OCs analysis was carried out using an Agilent 7890A/7000A GC triple quadrupole mass spectrometer system (GC/QQQ), and the gas chromatograph was equipped with the Agilent 7693 autosampler. Data analysis and quantification of individual OCs were conducted using Agilent MassHunter Workstation software—Quantitative Analysis. The fractions were injected in the splitless mode and the injector was kept at 270 °C. A (35% Phenyl)-methylpolysiloxane DB 35 ms capillary column (30 m × 0.25 mm id, 0.25 mm film thickness) (J&W Scientific from Agilent, Santa Clara, CA, USA) was used for analytes separation and helium was employed as carrier gas. The oven temperature was programmed as follows: hold for 1 min at 120 °C, increase 15 °C min^−1^ to 160 °C, increase 3 °C min^−1^ to 220 °C, hold for 2 min at 220 °C, increase 8 °C min^−1^ to 280 °C, hold for 4 min at 280 °C. The GC/MS/MS operated in multiple reaction monitoring (MRM). The QQQ collision cell helium quench gas was set to 2.25 mL min^−1^ with N2 collision gas at 1.5 mL min^−1^. Two MRM transitions were used for each OC to quantify and qualify the compounds. The calibration curves were linear with a R^2^ coefficient greater than 0.998.

For quality assurance and quality control, we mainly followed the guidelines proposed by the European Commission Regulation No. 644/2017. To confirm method selectivity and to check for cross-contamination, matrix, and procedural blanks were extracted and analyzed. Pure reference standard solutions (Dr. Ehrenstorfer Laboratory) were used for instrument calibration and analytes quantification, as well as for precision and recovery tests. Recovery rates, calculated at 3 concentration levels, ranged from 81% to 105% depending on the compound. Analytical performances of the method were also assured by the analyses of reference materials: IAEA-406 for OCPs and ndl-PCBs and WMF-01 for dl-PCBs (International Atomic Energy Agency and Wellington Laboratories, respectively). Since the GC-MS/MS produces virtually no noise in the chromatogram, the standard deviation (SD) of the lowest reference standard solution in seven replicates was calculated and used to determine the limit of detection (LOD) and limit of quantification (LOQ) values (assumed as 3 × SD and 10 × SD, respectively). They ranged from 0.005 ng mL^−1^ to 0.03 ng mL^−1^ and from 0.02 ng mL^−1^ to 0.10 ng mL^−1^ (on solvent), respectively, for detection and quantification limits. Data, not corrected for recovery, were expressed as ng g^−1^ on lipid weight (LW), as well as on wet weight (WW). and they were reported as not detectable (ND) when below LOD. For the calculation of sum and mean values, we chose to assume a value of zero for concentrations lower than LOD.

### 2.4. Parasitological Examination

Skin, gills, mouth cavity, GI tract (stomach and intestine), liver, heart, gonads, visceral cavity, and mesenteries of each fish were dissected and studied for metazoan parasites using a Zeiss Axio Zoom V16 (Carl Zeiss, DE) dissecting microscope [42,43]. The musculature of each specimen was cut into thin slices (~0.5 cm) and examined under the same microscope for helminth parasites [44]. For each organ/tissue, ecto- and endo-parasites were studied, collected, counted, washed in physiological saline solution, and preserved in 70% ethanol. For morphological identification, trematodes and acanthocephalans were stained with Mayer’s acid carmine dehydrated through a graded ethanol series, cleared in methyl salicylate, and mounted in Canada balsam; nematode larvae were clarified in Amman’s lactophenol. All parasites were then studied by light microscope and identified to the lowest possible taxonomic level using the available literature. Parasites were found almost exclusively in the GI (mostly in the intestine). For this reason, we referred to GI parasites throughout the text.

### 2.5. Statistical Analysis

Statistical analysis was performed using STATA software. Data obtained from OC determinations and from parasite detection were preliminarily evaluated through summary statistics and presented as mean ± SD. To assess and quantify the relationship between parasitic infection and OC contamination levels, regression analysis was exploited. The outcome variable was either a dummy identifying sample units characterized by the presence of parasites or the variable measuring the (average) number of parasites; the right side of the regression contains a measure of OC compounds divided by the (average) length of the trout. In this latter case, since the outcome variable is a count variable that takes on nonnegative integer values, we considered a Poisson regression model, which is well suited for modeling count variables. Moreover, we also provided evidence by adding two control variables, that is sex and sexual stage, to check whether differences in these parameters can be responsible for a confounding effect on the results of the previous analysis.

Measures of OCs concentrations were the sum of OCs concentrations on LW, considering the following groups of compounds: Σ6ndl-PCBs, Σ15ndl-PCBs, Σ3dl-PCBs, Σ5dl-PCBs, ΣDDTs, HCB and dieldrin. The null hypothesis (absence of a relationship between the two variables of interest) was rejected in the case of a *p*-value lower than 0.05.

## 3. Results and Discussion

Fork length ranged from 10.65 to 20.1 cm (mean value: 14.18, SD: 2.49); the weight ranged from 18.9 to 151.3 g (mean value: 57.18, SD: 29). Among the 22 sample units assembled, 11 units consisted of only female trout and 11 of only male ones; 45% of female sample units and 27% of male sample units were sexually mature. The sample mean fat content was 0.033 g g^−1^ muscle (range 0.080–0.130 g). GI parasites infested 54% of sample units.

### 3.1. OCs in Trout Muscle

PCBs were the most plentiful OCs followed by DDTs, HCB, and dieldrin (Table 1 and Table 2). Among PCBs, the mean concentration was highest for the Σ15ndl-PCBs group, followed by Σ6ndl-PCBs, Σ5dl-PCBs, and Σ3dl-PCBs (Table 1). The Σ6ndl-PCBs amounted to roughly 72% of Σ15ndl-PCBs.

Each indicator ndl-PCB congener was detected in all sample units (Figure 2). The indicator hexa-chlorinated PCBs IUPAC Nos. 153 and 138 showed the highest mean concentrations among all OCs, with values of 63.25 and 49.74 ng g^−1^ LW, respectively. Adding up these two congeners, we accounted for 56% and about 40% of Σ6ndl-PCBs and Σ15ndl-PCBs, respectively. The tri-chlorinated PCB 28 followed with a mean concentration of 33.77 ng g^−1^ LW. The indicator tetra-chlorinated PCB 52 was detected at the lowest concentration, contributing to 7% of Σ6ndl-PCBs (Figure 2).

Regarding the non-indicator ndl-PCBs, the tetra-chlorinated PCB 66 was detected with the highest mean concentration (21.56 ng g^−1^ LW), accounting for 7.6% of Σ15ndl-PCBs, while the PCBs Nos. 194 and 196 were detected with the lowest ones (1.30 and 1.56 ng g^−1^ LW, respectively). The latter two congeners were found in 65% of sample units; other non-indicator ndl congeners were in all sample units.

Regarding dl-PCBs, Σ5dl-PCBs were higher than Σ3dl-PCBs (mean value of 35.44 ng g^−1^ LW vs. 11.30 ng g^−1^ LW) (Table 1). Among non-ortho congeners, PCBs Nos. 126 and 169 were never detected (Figure 3); therefore, the concentration of Σ3dl-PCBs is only due to PCB 77 (mean value of 11.28 ng g^−1^ LW and detection in 83% of sample units). Four out of five mono-ortho PCBs were detected; in particular, in all sample units, we found PCB 105 and PCB 118 (mean values 7.99 and 21.96 ng g^−1^ LW, respectively). The latter congener showed the highest concentration among dl-PCBs and contributed 62% to Σ5dl-PCBs. PCB 156 and PCB 167 had relatively low mean concentration levels (3.63 and 1.86 ng g^−1^ LW) (Figure 3); they were detected in 69% and 78% of the sample units, respectively. The congeners PCB 118, PCB 77, and PCB 105, recognized as more toxic and usually detected at lower concentrations than ndl ones [45], were instead found at mean concentrations roughly comparable to those of some indicator and non-indicator ndl-PCBs (i.e., PCB 52, 101, 180, 66, 74).

The finding of relatively high concentrations of PCBs fairly reflects our expectations. In fact, a National Interest Site and three sites contaminated with OCs undergoing remediation procedures are located in the same province of the study area.

Regarding OCPs, *p,p’*-DDE and HCB were found in all sample units with the highest mean concentrations (97.57 and 8.09 ng g^−1^ LW, respectively); *p,p’*-DDE was the dominant compound in terms of contribution to ΣDDTs (97%) (Table 2). *p,p’*-DDD, *p,p’*-DDT, and dieldrin were detected in 83%, 26%, and 43% of sample units, respectively, with quite low concentrations (Figure 4).

As it is well known, DDT is quickly metabolized to DDE that persists for a long time in the environment. Indeed, the relatively high *p,p’*-DDE concentration suggests that the parent compound was strongly used in the past. This hypothesis is supported by the value of the DDE/DDT ratio, based on the average concentration of the compounds, which is usually used to investigate the timing of DDT entering the environment [46]. In our case, the ratio is much higher than 1 (amounting to 123). The reasons behind the presence of HCB can be identified both with its past use as a fungicide and because it is a by-product of several chlorine-containing chemicals [47].

Trout feeding could explain the presence of OCs. Crustaceans, insects, and small fish are the main components of trout’s diet. These organisms live on the seabed in direct contact with sediments that accumulate OCs.

### 3.2. Regression Analysis of Parasites Infection and OCs

A total of 233 helminth parasites belonging to three taxa were found in the GI tract of 41 brown trout. The most prevalent (42.5%) and abundant taxon (mean abundance: 5.4 per specimen ranging from 1 to 35) was the trematode *Crepidostomum metoecus*, with the other trophically transmitted parasites found in 5.7% (the acanthocephalan *Echinorhynchus* sp.) and 2.3% (unidentified nematode larva) trout, respectively (Table 3). *Crepidostomum metoecus* has been previously reported as a common parasite of brown trout in European countries [31]. However, its presence has been rarely investigated in Italian rivers [48].

The results of the regression analysis are shown in Table 4, Table 5 and Table 6. When the outcome variable identified the presence or not of parasite infection, that is, the outcome variable was a dummy, and OC concentration levels were expressed on LW basis, the relationship between OCs and parasite presence was (i) negative and statistically significant in the case of Σ6ndl-PCBs, Σ15ndl-PCBs, Σ5dl-PCBs, and ΣDDTs; and (ii) negative but insignificant in the case of Σ3dl-PCBs, HCB, and dieldrin (see Table 4). These results suggest that relatively high OC concentrations might reduce the probability of parasite infection.

The use of the two control variables enabled us to verify if previous results were biased by potential confounding factors. Table 5 shows that the negative and statistically significant relationship between parasite infection and OC concentration still emerged for Σ6ndl-PCBs, Σ15ndl-PCBs, and Σ5dl-PCBs. Moreover, we also notice that while the variable sex does not have any statistical impact, the coefficient attached to the stage of sexual development (immature versus mature) was estimated negative for all regressions, and statistically significant in several cases, consistent with the idea that immature younger fish had less time to accumulate parasite.

Our main evidence was also confirmed by exploiting the Poisson regression analysis where the outcome variable was the number of parasites (see Table 6). A negative and statistically significant relationship between parasite infection and OC concentration was estimated for Σ6ndl-PCBs, Σ15ndl-PCBs, and Σ5dl-PCBs.

Our evidence regarding the negative relationship between chemical pollutants and parasites accords with that of other authors. By relying on a meta-analysis of the effects of pollution on parasitism in aquatic animals, Blanar et al. [26] conclude with a strong, significant negative effect for Digenea and Monogenea, especially in response to metal pollution. For many other parasite/contaminant interactions, effect sizes were instead not significantly different from zero. However, few studies focus on OCs and do not reach a consensus [25,49].

The digenetic class of trematodes, to which *Crepidostomum metoecus* belongs, has been reported as more vulnerable than other parasite species and a negative correlation with OCs has been noticed [26]; for freshwater taxa, the correlation was not determined due to lack of data relative to OC concentration levels. Vidal-Martinez et al. [50] evidenced a significant negative correlation between the degree of parasitism by larval trematode *Mesostephanus appendiculatoides* and DDT concentrations in the Mayan catfish collected in Chetumal Bay, Mexico. The authors reported that, although the catfish were highly contaminated, they seemed to be barely susceptible to the OCPs toxicity, which instead heavily affected parasites. A negative correlation was also documented with respect to the digenetic trematode *Steringophorus furciger* in the GI tract of flatfish species, which is a significantly greater presence of parasites in specimens poorly contaminated by PCBs [51]. Similar evidence was also reported by Carreras-Aubets et al. [25], who observed a lower abundance of adult digenean endoparasite *Opecoeloides furcatus* in *Mullus barbatus* fish related to higher concentrations of the sum of indicator PCBs plus PCB 118 in Western Mediterranean sediments. To the best of our knowledge, a positive relationship has never been recorded for the digenetic class of trematodes. With data from German Bight, Schmidt et al. [52] evidenced a negative relationship between the infection degree to some parasite species and OC concentrations measured in the muscle of European flounder, sediments, and blue mussels. In particular, they documented a lower abundance of parasites with higher concentrations of HCB, DDD, DDE, and the sum of indicator PCBs. In the liver and muscle of acanthocephalan-infected perch, Brázová and co-authors [53] detected PCB concentrations several times lower than in not parasitized perc. Similar findings, observed only for some PCB congeners, were recently reported by the same authors for another host-parasite system (GI cestode–freshwater bream) [54].

In addition to aquatic species, a negative correlation was also evidenced in dogs [55] and African immigrants [24]. The authors showed that serum concentration levels of some indicator PCBs (PCB 52, PCB 138, PCB 153) were significantly lower in subjects infected with GI parasites than in non-parasitized ones. The same results were found examining several OCs, including those that we analyzed in the present study, in dogs positive for earthworm *Dirofilaria immitis* [55]. Since OC serum concentrations remained low after antiparasitic treatment, the authors hypothesized that nematode parasites may metabolize OCs in addition to accumulating them.

Some other studies reported, instead, an increase in parasitism because of animal exposure to OCs. Parasitic infection prevalently by nematodes of the lung (but also intestine and uterus) was significantly and positively correlated to *p,p’*-DDT, *p,p’*-DDD, and *o,p’*-DDD concentrations measured in the blubber of finless porpoises (*Neophocaena phocaenoides*) [56]. In blubber of the same species, Isobe et al. [49] evidenced significantly higher concentrations of PCBs (expressed as the sum of 62 congeners including indicator congeners and dl-PCBs) in animals infected with liver trematodes, relative to non-infected ones. A similar correlation was also reported by Kannan and co-authors [57] for PCBs and infectious agents including intestinal acanthocephalans, protozoa, bacteria, and fungi in another marine mammal, the Southern Sea otter. Bamidele and co-authors [58,59] observed a significant positive correlation between the protozoa *Myxosoma sp* infection found in blackchin tilapia and silver catfish from Lagos lagoon (Nigeria) and *p,p’*-DDT concentrations in freshwaters; however, no correlation was determined between intestinal helminth parasites and OCP congeners [58].

Few potential explanations have been raised for the results obtained. Some authors related the findings of a positive relationship to the frequent immunosuppressive effect of the OCs reported in *in vivo* and *in vitro* studies [6,60,61,62]. Indeed, exposure to chemical xenobiotics can disrupt the host’s immune response, so that adverse effects may depend on the indirect outcome of the pollutant on the host’s capacity to cope with a pathogen [13], making the host itself more susceptible to parasitosis [63,64]. Instead, regarding the negative correlation, some authors suggest that high environmental OC concentrations may disrupt the parasite life cycle, impairing the ability of the free-life form to survive and reach the host [25,51,65]. To protect themselves from chemical contaminants, parasites may rely on their host for detoxification mechanisms, even though the parasitic infection itself could cause a decrease in host detoxification enzymes [65]. In the case of trematodes, which have a complex life cycle with a first intermediate host, usually a mollusk, several free-living forms, and, finally, one or multiple vertebrate hosts, the toxic effects of the pollutants may impair their ability to assimilate essential nutrients; as a consequence of this condition, also their reproductive potential results to be affected and some mutagenic damages could occur to developing embryos, leading to a reduced transmission [65].

The negative relationship has been also interpreted suggesting that parasites determine a decrease in OC concentrations. The explanation would be that, because of their lipid content, GI parasites, and thus also trematodes, are able to bioaccumulate OCs introduced by the animal host through contaminated feed consumption [21,53,54,66,67]. This makes OCs less bioavailable for the host and therefore may increase the tolerance of the host to contaminants-induced toxicity. For parasites other than helminths, another explanation may result from their metabolizing ability through cytochrome P-450 activity leading to the reduction of OCs body burden in the host [68]. Indeed, helminths perform limited OC detoxification because of the lack of cytochrome P450 mono-oxygenases [69,70]. Moreover, according to Brázová and co-authors [54], since a higher Fulton’s condition factor was determined in infected breams, some parasites may even positively impact their hosts co-exposed to PCBs.

Overall, assessing the potential outcomes for parasites exploiting contaminated hosts is a crucial but neglected issue, since toxic effects on parasites may alter interspecific relationships. However, joint effects of parasites and chemical pollutants on host performance are extremely intricate depending on the level of parasitism, as well as the chemical pollutant, its mode of action, and the exposure levels. Contaminant accumulation by parasite results in positive effects among which is a reduction of both oxidative stress and histological modifications. This scenario indicates the possibility that a shift from parasitism to mutualism occurs. If concentration levels of the contaminant are high, they could induce damage to both host and parasites changing their relationships (see [71] and references therein).

### 3.3. Risk Assessment

Human exposure to OCs is due to a large extent to the consumption of foods of animal origin, mainly dairy and fishery products [72]. For consumer health risk assessment, the European Union (EU) established maximum residue limits (MRLs) for some OCs, namely dioxins, dl-PCBs, and six indicator ndl-PCBs in foodstuffs of animal origin (European Commission Regulation no. 1259/2011). The regulation fixed for “Muscle meat of wild caught freshwater fish, with the exception of diadromous fish species caught in freshwater, and products thereof” a MRL of 125 ng g^−1^ WW for ∑6ndl-PCBs. All the samples analyzed showed concentrations below this threshold.

For dioxins and dl-PCBs, MRLs are expressed as Toxic Equivalent Quantity (WHO-TEQ). To evaluate the compliance of any sample units with the thresholds set by the EU, we summed up the concentration of each congener of interest multiplied by the respective Toxic Equivalency Factor adopted by the World Health Organization (WHO-TEFs) and then compared the sum with the corresponding MRL. TEFs are the results of the comparison of the relative different toxicity of individual dioxins/furans and dl-PCBs congeners to that of 2,3,7,8-TCDD [73]. Since there is no specific limit for dl-PCBs, we considered a value of 3 pg g^−1^ fat arising from the difference between the MRL referred to as the sum of all dioxin-related compounds (PCDDs/Fs and dl-PCBs) and the value referred to PCDDs/Fs alone. Also, in this case, the measured concentrations were lower than the above-extrapolated value.

With respect to the considered pesticides, the EU has not set a specific MRL for fish and fish products to date. As far as concern Italian national legislation, tolerance limits for residues of chloro-organic substances in products of animal origin have been established by Decreto Ministeriale on 13 May 2005 (annex 4), which states limits differentiated according to the different fat content. All sample units in the present study showed a lipid percentage below 5%, thus falling into the so-called group 1, with tolerance limits for ∑DDTs, HCB, and Dieldrin, respectively, of 0.050, 0.010, and 0.005 µg g^−1^ WW. No sample units exceeded the limits of the above-mentioned pesticides established by the Italian legislation.

## 4. Conclusions

We investigated the potential existence of a relationship between parasitic infection degree and OC concentration levels found in brown trout specimens from the Calabria region. The regression analysis revealed the possibility that relatively high PCB concentrations lead to lower parasite infection degree. Results supporting this conjecture are summarized as follows: (i) there is a strongly significant negative relationship between parasites and PCB concentrations; (ii) such relationship is robust to the inclusion of two regressors in the empirical model that controls for sex and sexual maturity of trout, although the magnitude of the significance reduces due to the relatively low number of observations; (iii) the negative relationship is confirmed by results of the Poisson regression model for counting data; and (iv) among all considered OCs, concentrations of PCBs, that is, Σ15ndl-PCBs, Σ6ndl-PCBs and Σ5dl-PCBs, are the highest ones.

However, we are aware that the sample size may be an issue in our analysis and that the reverse causal direction cannot be dismissed. Therefore, more studies about the relationship between chemical pollutants and parasites in host–parasite systems, assessing whether a concern would arise for the host, the parasite, or the final consumers would be beneficial.

While OC concentration levels detected do not pose a risk to consumer health, being always below the threshold levels set by the EU, they signal non-negligible background pollution by PCBs and OCPs, although the monitored area is a protected one.

## Figures and Tables

**Figure 1 animals-13-02961-f001:**
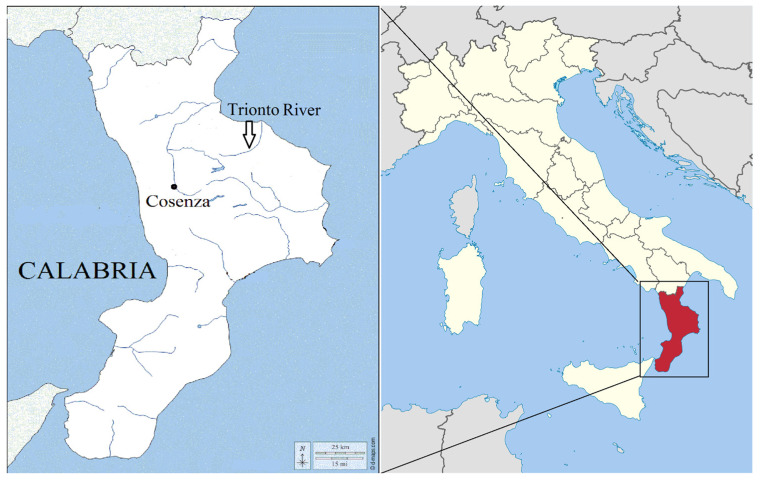
Italy and sampling area in Calabria Region.

**Figure 2 animals-13-02961-f002:**
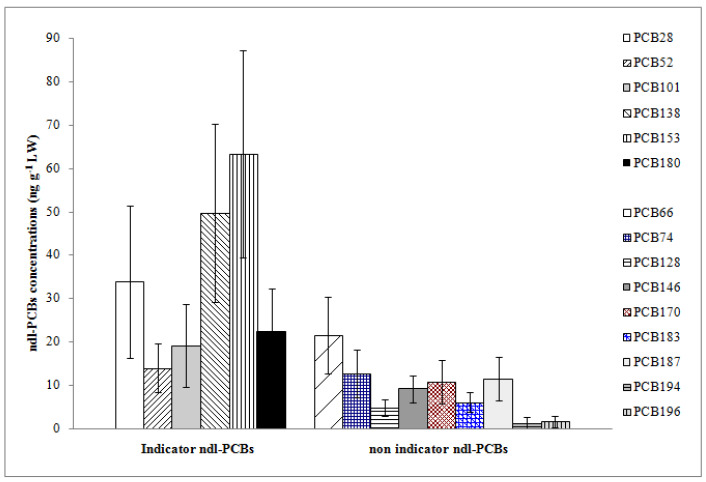
Mean concentrations and standard deviation of indicator and non-indicator ndl-PCBs expressed as ng g^−1^ on lipid weight basis (LW).

**Figure 3 animals-13-02961-f003:**
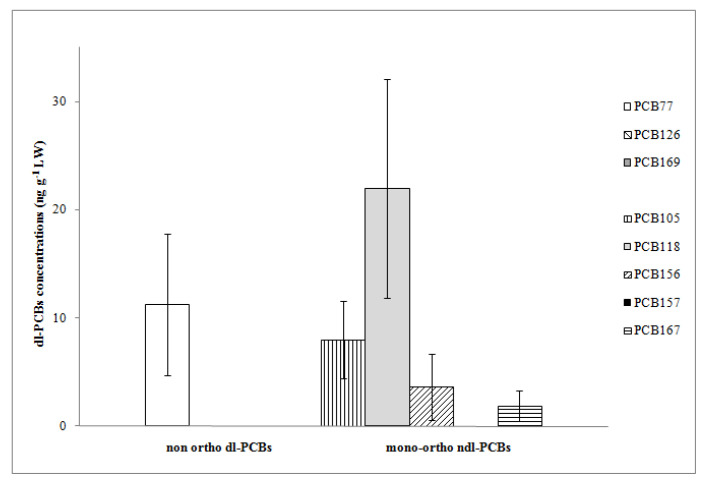
Mean concentrations and standard deviation of dl-PCBs expressed as ng g^−1^ on lipid weight basis (LW).

**Figure 4 animals-13-02961-f004:**
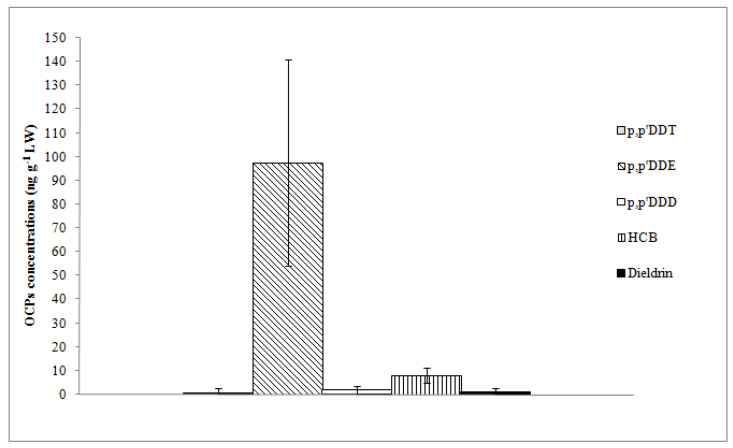
Mean concentrations and standard deviation of OCPs expressed as ng g^−1^ on lipid weight basis (LW).

**Table 1 animals-13-02961-t001:** Summary statistics of ndl-PCB and dl-PCB concentrations (ng g^−1^).

	Mean	SD	Min	Max
Panel A (LW)				
Σ6ndl-PCBs	201.99	80.69	97.82	408.04
Σ15ndl-PCBs	281.18	111.23	140.03	560.86
Σ3dl-PCBs	11.3	6.53	0.00	22.04
Σ5dl-PCBs	35.44	16.85	14.61	79.79
Panel B (WW)				
Σ6ndl-PCBs	6.58	2.55	3.61	13.60
Σ15ndl-PCBs	9.15	3.49	5.10	18.69
Σ3dl-PCBs	0.36	0.21	0.00	0.74
Σ5dl-PCBs	1.16	0.55	0.49	2.66

Notes: The table reports PCB concentrations expressed as ng g^−1^ on lipid weight basis (LW, Panel A) and wet weight basis (WW, Panel B). Σ6ndl-PCBs refers to the sum of the concentrations of six indicator PCBs; Σ15ndl-PCBs indicate the sum of indicator and non-indicator ndl- congeners.; Σ3dl-PCBs and Σ5dl-PCBs are the sums of concentrations of 3 non-ortho and 5 mono-ortho congeners, respectively.

**Table 2 animals-13-02961-t002:** Summary statistics of OCP concentrations (ng g^−1^).

	Mean	SD	Min	Max
Panel A (LW)				
HCB	8.09	3.11	2.63	13.97
Dieldrin	1.25	1.60	0.00	5.09
*p,p’*-DDT	0.79	1.78	0.00	7.51
*p,p’*-DDD	1.86	1.94	0.00	8.60
*p,p’*-DDE	97.57	43.45	38.33	228.42
ΣDDTs	100.23	43.69	42.72	228.42
Panel B (WW)				
HCB	0.27	0.11	0.10	0.49
Dieldrin	0.04	0.05	0.00	0.14
*p,p’*-DDT	0.02	0.05	0.00	0.20
*p,p’*-DDD	0.06	0.06	0.00	0.23
*p,p’*-DDE	3.19	1.46	1.42	7.61
ΣDDTs	3.28	1.45	1.43	7.61

Notes: The table reports OCP concentrations expressed as ng g^−1^ on lipid weight and wet weight basis (Panel A and Panel B, respectively). ΣDDTs refers to the sum of the three DDT-related compounds.

**Table 3 animals-13-02961-t003:** Taxa and number of gastro-intestinal (GI) parasites found in the infected trout.

	Number of Parasites			Number of
	Total	Min	Max	Infected Trout
*Crepidostomum metoecus*	225	1	35	37
*Echinorhynchus*	4	1	1	4
*Unidentified nematodes*	2	1	1	2
Other unidentified parasites	2	1	1	2
All	233	1	35	41

Notes: some trout hosted more than one species of parasites.

**Table 4 animals-13-02961-t004:** Regression analysis: presence of parasites and OCs.

Presence of Parasites							
Σ6ndl-PCBs	−0.034 ***						
	(−4.69)						
Σ15ndl-PCBs		−0.024 ***					
		(−4.76)					
Σ3dl-PCBs			−0.350				
			(−2.03)				
Σ5dl-PCBs				−0.154 ***			
				(−4.04)			
ΣDDTs					−0.055 **		
					(−3.54)		
HCB						−0.633	
						(−2.02)	
Dieldrin							−0.837
							(−0.14)

Notes: The table reports results of the regression analysis of GI parasites on OC concentration levels (LW); t statistics in brackets. Number of observations: 22. Significance is denoted as follows: ** *p* < 0.01, *** *p* < 0.001.

**Table 5 animals-13-02961-t005:** Regression analysis: presence of parasites and OCs, with control variables.

Presence of Parasites							
Σ6ndl-PCBs	−0.024 *						
	(−2.49)						
Σ15ndl-PCBs		−0.018 *					
		(−2.49)					
Σ3dl-PCBs			−0.211				
			(−1.12)				
Σ5dl-PCBs				−0.115 *			
				(−2.37)			
ΣDDT					−0.035		
					(−2.00)		
HCB						−0.175	
						(−0.51)	
Dieldrin							−5.771
							(−1.50)
Immature	−0.395	−0.395	−0.463 *	−0.436 *	−0.422	−0.494 *	−0.601 **
	(−1.91)	(−1.90)	(−2.28)	(−2.21)	(−1.98)	(−2.63)	(−3.70)
Male	0.088	0.089	0.149	0.096	0.122	0.139	0.197
	(0.48)	(0.48)	(0.79)	(0.52)	(0.64)	(0.70)	(1.03)

Notes: the table reports results of the regression analysis of GI parasites on OC concentration levels (LW) controlling for sex and sexual maturity; t statistics in brackets. Number of observations: 22. Significance is denoted as follows: * *p* < 0.05, ** *p* < 0.01.

**Table 6 animals-13-02961-t006:** Poisson regression analysis.

Number of Parasites							
Σ6ndl-PCBs	−0.079 **						
	(−2.59)						
Σ15ndl-PCBs		−0.056 *					
		(−2.55)					
Σ3dl-PCBs			−0.464				
			(−1.06)				
Σ5dl-PCBs				−0.439 *			
				(−2.09)			
ΣDDT					−0.104		
					(−1.43)		
HCB						−1.177	
						(−1.24)	
Dieldrin							−8.940
							(−0.74)
Immature	−1.054	−1.054	−1.181 *	−1.160 *	−1.072	−1.091	−1.398 **
	(−1.90)	(−1.91)	(−2.30)	(−2.10)	(−1.80)	(−1.79)	(−2.69)
Male	0.561	0.558	0.555	0.628	0.555	0.522	0.546
	(1.39)	(1.39)	(1.35)	(1.40)	(1.28)	(1.27)	(1.21)

Notes: The table reports results of the Poisson regression analysis of the number of GI parasites on OC concentration levels (LW) controlling for sex and sexual maturity; t statistics in brackets. Number of observations: 22. Significance is denoted as follows: * *p* < 0.05, ** *p* < 0.01.

## Data Availability

The data supporting the findings of the present study are available from the corresponding author upon reasonable request.

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
