# Peer review of "Do Organochlorine Contaminants Modulate the Parasitic Infection Degree in Mediterranean Trout (Salmo trutta)?"

_animals, 2023, doi:10.3390/ani13182961_

Round 1

Reviewer 1 Report

The authors investigated the effect of organochlorine contaminants on parasitic infection degree in Mediterranean trout. This manuscript (MS) was not clearly written and easy to understand. They did not cover enough range of factors and still some gaps are here in terms of provided data. This work could help the sustainability of this species farming if more data could be provided and written better. However, some major issues significantly compromised the quality of this MS.

Major comments:

  • First, the manuscript needs to be edited by a native English speaker to improve the language of the MS and fix errors.
  • They did not provide adequate data and evidence to support their hypothesis.

·       The writing style is so confusing and hard to understand.

·       Due to these problems, I did not go through line-by-line comments.

Best regards

NA

Author Response

Dear Reviewer

our investigation revealed the possibility that high PCB concentrations lead to lower parasite infection degree.

We are aware that, as for other studies, the sample size may be an issue. However, results supporting our conclusion can be summarized as follows: (i) among all considered OCs, chromatographic analysis revealed higher  concentrations for ndl-PCBs (Σ15ndl-PCBs and Σ6ndl-PCBs); (ii) at the same time, correlation analysis revealed a strongly significant negative relationship between parasites and ndl-PCBs; (iii) such relationship remains when we control for two  potential confounding factors, that is sex and sexual maturity of trout; (iv) the negative relationship is confirmed by results of the Poisson regression model for counting data.

Moreover notice that, before to submit the new version of the paper, we have checked the main text for grammatical errors and rephrased some sentences poorly written.

Reviewer 2 Report

Important type of work. Interesting results. Please note that the results cannot imply causation or causal relationship, as the R values are very low, i.e. very weak correlations in most of the results.

Important type of work. Interesting results. Please note that the results cannot imply causation or causal relationship, as the R values are very low, i.e. very weak correlations in most of the results. Please see comments and questions in review and respond.

Author Response

We thank the reviewer for the useful suggestions aimed at improving our research article. We took into account all his/her comments. The reply point-by-point to the reviewers’ comments (yellow highlighted in the text) is reported below.

REVIEWER 2

General Comments and Suggestions for Authors

Important type of work. Interesting results. Please note that the results cannot imply causation or causal relationship, as the R values are very low, i.e. very weak correlations in most of the results. Please see comments and questions in review and respond

Simple Summary

Line 26: replace Intestinal with intestinal (lowercase letter)

Done

Abstract

Line 44: replace always with consistently

Done

Introduction

Line 56: replace few with some

Done

Line 57: replace answer with biological response

Done

Results and Discussion - 3.2 Regression analysis of parasites infection and OCs

Can you provide a table to summarise all the parasites found?

A new table (Table 3), summarizing the number and taxa of parasites has been added.

Line 316 (old Table 3, new Table 4): Most of these correlations are very weak (R < 0.25). How significant is the causal relationship between pesticide and contaminant concentration and the load of parasites?

Line 328 (old Table 4, new Table 5): Most of these correlations are very weak (R < 0.25). How significant is the causal relationship between pesticide and contaminant concentration and the load of parasites? How is this R value > 1?

Line 337 (old Table 5, new Table 6): How is this R value > -1 ?

We are aware that we cannot be assertive about the causal relationship, as recognized in the concluding section. On the basis of all our results, we only conjecture the possibility of a potential negative causal relationship as suggested by other authors cited in the main text:

1) Concentrations of PCBs, that is Σ15ndl-PCBs, Σ6ndl-PCBs, Σ5dl-PCBs are the highest among those of all considered OCs.

2) There is a strong statistical relationship (t>4) between parasites and PCB concentrations as reported in the new Table 4;

3) The statistical significance is robust to the inclusion of two control variables, although the magnitude of the significance lowers (not surprisingly given the relatively low number of observations) and is confirmed by results of the Poisson regressions for counting data.

Line 350: Replace Digenic with digenetic

Done.

Lines 351-352: Organochlorines are pesticides aimed at controlling parasites. So one would expect that residue OC in the tissue of fish may suppress the development of sensitive parasite species. However, it is also well known that parasites develop resistance to pesticides. Therefore, the negative correlation or negative relationship between dose and parasite load can change.

Regression analysis evidenced that the relationship between parasite load and OCPs is statistically insignificant. However, note that values of OCP concentrations are quite low, perhaps not enough to determine a statistical relationship.

Lines 361-363: PCBs are not designed to kill parasites. If a fish has lots of PCBs and its parasite load is low, the mechanism may be the induction of oxidative stress (production of oxygen radicals) and this process may damage parasites that live in the fish tissues undergoing high levels of ROS production.

We totally agree with the reviewer. PCB induce oxidative stress in different experimental models as observed by several researchers (see for example Liu J, Tan Y, Song E, Song Y. A Critical Review of Polychlorinated Biphenyls Metabolism, Metabolites, and Their Correlation with Oxidative Stress. Chem Res Toxicol. 2020 33(8):2022-2042. doi: 10.1021/acs.chemrestox.0c00078 and references therein). It is plausible to speculate that PCBs cause oxidative stress also in the parasite in addition to the host, damaging them. However, at the best of our knowledge no study has analyzed this issue.

Lines 374-376: The parasites may be a partition compartment for the PCBs as they can have lipids in their tissues, and store up the PCBs. This makes the PCBs less bioavailable for the fish host, and reduces toxicity to the fish?

Yes, we agree with the reviewer. The explanation has been given by some authors who evidenced that parasites are able to store up the OCs introduced by the animal host through contaminated feed consumption (as reported in the manuscript at lines 430-432 of the original manuscript). Discussion is present in the manuscript.  

Lines 394-396: Positive correlations of parasites to pollutant contaminants are probably related to immunosuppression by the pollutant on the host.

We totally agree with the reviewer. As reported in the manuscript (lines 405-410 of the original manuscript, lines 413-417 of the revised version) the OCs may induce an immunosuppressive effect as reported in several in vivo and in vitro studies. Indeed, exposure to pollutants can disrupt host immune response, so that adverse effects may depend on the indirect outcome of the pollutant on the host capacity to cope with a pathogen, making the host itself more susceptible to parasitosis.

Reviewer 3 Report

This paper describes some interesting host-parasite interactions along with the dual stressor of exposure to and bioaccumulation of OCs in the wild. This is a very interesting and relevant topic, and there are few places in the world where these types of dual stressors can be reliably studied. I think the information will be of interest to the readers of Animals, but I several specific comments that could help improve the paper for this purpose.

Specific comments:

Line 3: lower case "Trutta".

Lines 22-24 (and throughout): Trout is typically used as a plural unless speaking about an individual fish (e.g., trout are, they are, etc.).

Line 26: lower case "intestinal"

Lines 35-36: All of these are forms of OCs? It is a little unclear at this point. 

Line 36: define LW.

Lines 42-43: Is it a good thing that the parasitic infection is suppressed, or are the parasites an important part of brown trout function? Give the reader some brief information (a concluding sentence) about what your findings mean or why they are important.

Abstract (overall): there are a lot of acronyms that the reader is not yet familiar with. Please define upon first use.

Line 57: suggest "the extent of stimulus or suppression (delete "answer" and parenthetical statement).

Lines 61-64: To the detriment of the host? No effect? Positive effect?

Lines73-74: move up to previous paragraph so no single sentence paragraphs in introduction. 

Line 75: regarding "impair their success in reaching the host" - Is the reduction in parasite numbers therefore a concern for the parasite populations?

Line 86: please rearrange the words "deriving by eating trout fillet contaminated."

Introduction (overall): Need a better connection between what a potential reduction in parasites means for the host, parasite, or both. It is unclear what the importance of this reduction might be ecologically.

Line 97: regarding "were collected" - Were fish collected throughout the entire length of the river? Specific sites? If specific sites, were there expectations that there would be higher or lower concentrations of OCs given proximity to problem locations (e.g., the National Interest Site)?

Lines 116-120: Define what sexual developmental stages were observed or used.

Line 122: average "fork" length?

Lines 142-144: I don't completely follow this sentence. Please rewrite for clarity.

Line 176: superscript "R2".

Lines 196-206: Were the parasites collected from all organs included in the analysis? If so, how? Were the prevalences examined by organ type? Most of the information in the abstract and results and discussion focuses on intestinal parasites. If that is the focus, and the other organs were not used, this should be specified in the methods and mention of other organ examinations removed to prevent confusion.

Lines 212-215: I do not fully understand what the dummy variable is or why it is in the analysis.

Line 226: Spell out fork length to begin the sentence.

Lines 228-229: 54% of each sex were infected, or 54% of the total number of fish examined were infected regardless of whether or not they were male or female? Also is this from the examination of all tissues, or just the intestines as reported in line 303?

Lines 234-237: Suggest removal of these lines and referencing the tables or figures parenthetically when specific information from these are mentioned in the results section.

Lines 239: define LW and WW in table notes.

Figure 2 (and others): can SD bars be shown below the mean also? Please define LW in figure caption.

Lines 266-269: I don't fully follow this sentence.

Lines 273-275": why is this noteworthy? Explain as part of the discussion.

Line 305: regarding "trophically transmitted" - Meaning transmitted through consumption of infested prey?

Lines 310-311: Still not sure what the dummy variable is or what role it plays in the analysis. Makes interpretation of the relationships difficult.

Tables 3, 4, and 5: the information in the notes should be up in the figure caption. The observation row should be removed given this never changes from 22 (just state that this came from 22 observations in the caption). What is the line below "Dieldrin" that separates the final regression coefficient from the t statistic?

Table 3: this information could be presented in a single column.

Line 324: regarding "didn't have any statistical impact" - agreed based on p-values, but do the numbers suggest that males have more parasites than females? Is this biologically significant?

Lines 324-325: regarding "sexual development" - immature versus mature?

Line 334: regarding "Poisson regression" and data shown in Table 5 - If the Poisson regression is thought to be needed (or was ultimately found to be needed), it should be your primary analysis/results. Moreover, this table shows all variables of consideration, and because sexual maturity had and effect, it should be included in the final results being presented. As such, the results in table 3 from which this covariate is missing seem invalid, as do the results in table 4 where a Poisson regression was not used. This stepwise process is okay for getting to the final results, but the information from only table 5 (the final step in the stepwise process) should be shown and interpreted in the manuscript. All previous conclusions drawn from tables 3 and 4 seem to be captured by the data presented in table 5.

 Line 343: regarding "parasite-metal interactions" - Are OCs a metal? If so, make this clear. If not, I don't understand the relevancy of this comparison.

Line 347: what is meant by relevance?

Line 351: regarding "more vulnerable" - than other classes of nematodes? Other parasite species?

Lines 356-358: This is interesting, and starts to speak to the importance of your findings.

Lines 349-376: If I understand this correctly, the parasite may be regulating the OC levels in the host due to the parasites susceptibility to OCs? Please summarize what all of this information means.

Lines 399-404: suggest removal of this paragraph. Terrestrial and aquatic host-parasite interactions do not occur in the same way. Keep the paper focused on aquatic organisms using the many other well cited literature on fish and aquatic findings.

Lines 411-431 are very interesting, and I like the opposing perspectives. The major thing I still think is missing is the effect of the parasite on the host. If no OCs are present and the parasite is found in high numbers in the host, do they have a negative, neutral, or positive effect on the host? Conversely, when parasites are reduced due to high OC levels, is this good or bad for the host?

Subsection 3.3: This is all good news. Are any of these chemicals still in use? Is there a potential risk of exceeding these levels in the future? How might the host-parasite interactions described herein affect this potential risk?

Lines 466-468: To what end? Concern for the host, the parasite, both, or the end consumer (humans)?

Line 471: regarding "the monitored area is a protected one" - Meaning that no humans are able to consume fish from that area?

The paper is understandable, but there are a lot of minor issues with use of the English language throughout, and in several places, whole sentences are difficult to understand. 

Author Response

We thank the reviewer for the useful suggestions aimed at improving our research article. We took into account all his/her comments. The reply point-by-point to the reviewers’ comments (green highlighted in the text) is reported below.

REVIEWER 3

Comments and Suggestions for Authors

This paper describes some interesting host-parasite interactions along with the dual stressor of exposure to and bioaccumulation of OCs in the wild. This is a very interesting and relevant topic, and there are few places in the world where these types of dual stressors can be reliably studied. I think the information will be of interest to the readers of Animals, but I several specific comments that could help improve the paper for this purpose.

Specific comments:

Line 3: lower case "Trutta".

Done

Lines 22-24 (and throughout): Trout is typically used as a plural unless speaking about an individual fish (e.g., trout are, they are, etc.).

Where possible, the word trout has been declined in the plural as suggested.

Line 26: lower case "intestinal"

Done

Lines 35-36: All of these are forms of OCs? It is a little unclear at this point.

Yes, they are all OCs. The sentence has been modified to make it more clear:

“The highest concentrations emerged for the sum of the 6 non-dioxin-like (ndl) indicator polychlorinated biphenyls (Σ6ndl-PCBs), followed by the 1,1,1-trichloro-2,2-di(4-chlorophenyl)-ethane (DDT), dioxin-like PCBs, hexachlorobenzene (HCB), and Dieldrin.” (lines 37-39).

.

Line 36: define LW.

Done

Lines 42-43: Is it a good thing that the parasitic infection is suppressed, or are the parasites an important part of brown trout function? Give the reader some brief information (a concluding sentence) about what your findings mean or why they are important.

A new sentence has been added as requested:

“The results evidenced the existence of interactions between the dual stressors in the host-parasite system in the wild.” (lines 45,46).

Abstract (overall): there are a lot of acronyms that the reader is not yet familiar with. Please define upon first use.

Done

Line 57: suggest "the extent of stimulus or suppression (delete "answer" and parenthetical statement).

Done

Lines 61-64: To the detriment of the host? No effect? Positive effect?

We have clarified this point. References have been added according to which parasites don’t cause significant damage to fish:

“Some authors suggest that these parasites don’t cause relevant damages to fish (Stephen A. Bullard and Robin M. Overstreet, Digeneans as Enemies of Fishes, chapter 14, Fish Diseases, Volume 2, 2008), just exploiting the intestinal fluids (Bartoli P, Boudouresque CF. Effect of the digenean parasites of fish on the fauna of Mediterranean lagoons. Parassitologia. 2007 Sep;49(3):111-720). (lines 70,71).

Lines 73-74: move up to previous paragraph so no single sentence paragraphs in introduction.

Done

Line 75: regarding "impair their success in reaching the host" - Is the reduction in parasite numbers therefore a concern for the parasite populations?

We have clarified this point (lines 79-81).

Line 86: please rearrange the words "deriving by eating trout fillet contaminated."

The sentence has been modified as follows: “We thus also evaluated the ensuing risk for human health due to consumption of contaminated fillet” (lines 105,106).

Introduction (overall): Need a better connection between what a potential reduction in parasites means for the host, parasite, or both. It is unclear what the importance of this reduction might be ecologically.

Lines 411-431 are very interesting, and I like the opposing perspectives. The major thing I still think is missing is the effect of the parasite on the host. If no OCs are present and the parasite is found in high numbers in the host, do they have a negative, neutral, or positive effect on the host? Conversely, when parasites are reduced due to high OC levels, is this good or bad for the host?

In order to ensure sexual reproduction, adult parasites don’t induce a relevant harm to the host. They mainly exploit the intestinal fluids. Therefore, parasites have a moderate negative impact on the host.

We do not have data for saying something about the connection between parasites and hosts. However, we have discussed this issue making clear the possibility that “parasites are able to perform a greater beneficial action in the host organism than the price of infection. The infected host exposed to moderate concentrations of chemical pollutants show decreased contamination levels, due to contaminants accumulation by parasites, resulting positive effects among which a reduction of both oxidative stress and histological modifications. This scenario indicates the possibility that a shift from parasitism to mutualism occurs. Joint effects of parasites and chemical pollutants on host performance are extremely intricate depending on the level of parasitism, as well as the chemical pollutant, its mode of action and the exposure levels. If concentration levels of the contaminant are high, they could induce damages to both host and parasites changing their relationships. In some cases, a parasite may be susceptible to pollutants toxicity showing decreasedinfection prevalence (Goutte et al., 2022 and references therein)”..

Line 97: regarding "were collected" - Were fish collected throughout the entire length of the river? Specific sites? If specific sites, were there expectations that there would be higher or lower concentrations of OCs given proximity to problem locations (e.g., the National Interest Site)?

The trout were collected in a well-defined area of the river, in the municipality of Longobucco, Cosenza province. In the Cosenza province there is a National Interest Site where there have been identified 20 contaminated sites undergoing remediation procedures. Three of them are contaminated with OCs. Therefore, the finding of relatively high concentrations of OCs in the sample fairly reflects our expectations.   

More details about the study area are included in the manuscript (line 98). A new sentence has been added in the “Results and Discussion” section to explain our expectations about OCs concentration levels (lines 281-283).

Lines 116-120: Define what sexual developmental stages were observed or used.

See lines 119-121:

“Then, slices were microscopically examined to evaluate sex and gonadal maturity stage (mature and immature trout), on the basis of criteria by Hajirezaee et al.”

Line 122: average "fork" length?

Yes, it was considered the average fork length. The specification has been added in the main text (lines 115,116).

Lines 142-144: I don't completely follow this sentence. Please rewrite for clarity.

The sentence has been rewritten as follows: “Regarding the OCs analysis, for the extraction procedure, including separation of the analytes from the lipid fraction, and the purification of the final extracts, we followed the method described by Ferrante et al.” (lines 150-153).

Line 176: superscript "R2".

Done

Lines 196-206: Were the parasites collected from all organs included in the analysis? If so, how? Were the prevalences examined by organ type? Most of the information in the abstract and results and discussion focuses on intestinal parasites. If that is the focus, and the other organs were not used, this should be specified in the methods and mention of other organ examinations removed to prevent confusion.

We have searched for the presence of parasites in several organs and tissues, but we found them almost exclusively in the digestive tract (mostly in the intestine). The method has been described in subsection 2.4  (Parasitological examination). For this reason, we referred to gastro-intestinal (GI) parasites throughout the manuscript.

A new sentence has been added to clarify this issue: “Parasites were found almost exclusively in the GI (mostly in the intestine). For this reason, we referred to intestinal parasites throughout the text.” (lines 214-216).

Lines 212-215: I do not fully understand what the dummy variable is or why it is in the analysis.

The dummy variable is a dichotomic 0/1 variable identifying with values equal to one sample units characterized by the presence of parasites, and 0 otherwise.

Line 226: Spell out fork length to begin the sentence.

Done

Lines 228-229: 54% of each sex were infected, or 54% of the total number of fish examined were infected regardless of whether or not they were male or female? Also is this from the examination of all tissues, or just the intestines as reported in line 303?

54% of the total sample was infested with gastro-intestinal parasites. As regards sexual stage 45% of female sample units and 27% of male sample units were sexually mature. We have clarified this issue in the manuscript (lines 238-240).

Lines 234-237: Suggest removal of these lines and referencing the tables or figures parenthetically when specific information from these are mentioned in the results section.

The lines have been deleted and we have referred to the tables and figures parenthetically when specific information from these have been mentioned in the manuscript.

Lines 239: define LW and WW in table notes.

Done

Figure 2 (and others): can SD bars be shown below the mean also? Please define LW in figure caption.

Done

Lines 266-269: I don't fully follow this sentence.

The sentence has been modified (lines 269-271).

Lines 273-275": why is this noteworthy? Explain as part of the discussion.

The sentence has been modified (lines 277-280).

Line 305: regarding "trophically transmitted" - Meaning transmitted through consumption of infested prey?

Yes, we refer to parasites that are transmitted through predator-prey feeding interactions in food webs.

Lines 310-311: Still not sure what the dummy variable is or what role it plays in the analysis. Makes interpretation of the relationships difficult.

The empirical analysis is based on two different regression models. The one with a dummy variable on the left side of the regression equation can be interpreted in terms of probability of parasite load determined by OCs concentration levels. Thus, a negative coefficient attached to a regressor implies that the probability of parasite presence reduces when PCB concentration increases.

Tables 3, 4, and 5: the information in the notes should be up in the figure caption. The observation row should be removed given this never changes from 22 (just state that this came from 22 observations in the caption). What is the line below "Dieldrin" that separates the final regression coefficient from the t statistic?

The tables have been modified as requested.

Line 324: regarding "didn't have any statistical impact" - agreed based on p-values, but do the numbers suggest that males have more parasites than females? Is this biologically significant?

The average number of parasites is 3.1 in males and 2.1 in females, but the difference is statistically insignificant.

Lines 324-325: regarding "sexual development" - immature versus mature?

Yes.

Line 334: regarding "Poisson regression" and data shown in Table 5 - If the Poisson regression is thought to be needed (or was ultimately found to be needed), it should be your primary analysis/results. Moreover, this table shows all variables of consideration, and because sexual maturity had and effect, it should be included in the final results being presented. As such, the results in table 3 from which this covariate is missing seem invalid, as do the results in table 4 where a Poisson regression was not used. This stepwise process is okay for getting to the final results, but the information from only table 5 (the final step in the stepwise process) should be shown and interpreted in the manuscript. All previous conclusions drawn from tables 3 and 4 seem to be captured by the data presented in table 5.

We agree with you that results reported in Table 5 (new Table 6) encompass those reported in previous tables. We would like to stress that our main evidence suggests that more PCB concentrations imply lower probability of parasite presence as well as lower number of parasites (Poisson regression).

 Line 343: regarding "parasite-metal interactions" - Are OCs a metal? If so, make this clear. If not, I don't understand the relevancy of this comparison.

The sentence has been modified (lines 360-365).

Line 347: what is meant by relevance? Line 351: regarding "more vulnerable" - than other classes of nematodes? Other parasite species?

We referred to other parasite species. Please, note that all the discussion has been modified.

Lines 349-376: If I understand this correctly, the parasite may be regulating the OC levels in the host due to the parasites susceptibility to OCs? Please summarize what all of this information means.

A brief conclusion has been added suggesting that OCs may be harmful for some parasites (line 441-443).

Lines 399-404: suggest removal of this paragraph. Terrestrial and aquatic host-parasite interactions do not occur in the same way. Keep the paper focused on aquatic organisms using the many other well cited literature on fish and aquatic findings.

Done

Subsection 3.3: This is all good news. Are any of these chemicals still in use? Is there a potential risk of exceeding these levels in the future? How might the host-parasite interactions described herein affect this potential risk?

All these chemicals have been banned in 2001 by the Stockholm Convention on Persistent Organic Pollutants. However, due to the high chemical stability and lipophilicity, they tend to persist in the environment for many years and to bioconcentrate and biomagnify in the food chains. As a consequence, OCs are still detected in abiotic and mainly biotic matrices of the ecosystems (see, for instance, Masset et al., 2019; Renieri et al., 2019) as well as in human samples (Barmpas et al., 2020). Therefore, in the next years we may still detect relatively high concentration levels of OCs in the environment, though the potential risk of exceeding current levels in the long-run is quite low. However, we notice that PCBs may also origin in case of illicit discharges and by-products of combustion processes and informal e-waste recycling that is increasing in Africa.

In our opinion the host-parasite interactions described in the manuscript don’t affect the potential risk for human health due to the consumption of contaminated fish.

Lines 466-468: To what end? Concern for the host, the parasite, both, or the end consumer (humans)?

Concern may be for the host, and the parasite and the balance of their relationship.

Line 471: regarding "the monitored area is a protected one" - Meaning that no humans are able to consume fish from that area?

In general, consumption of fish from that area is allowed. However, being a protected area, fishing activity is subject to authorization (Sila National Park) and restricted for the protection of the ecosystem.

Round 2

Reviewer 1 Report

They could not improve the quality of MS and still my decision is reject.

Required deep revision.

Reviewer 3 Report

The authors have done a good job of addressing all of the reviewer comments and the paper has greatly improved as a result. 

Overall, the quality of the English language is good enough for readers to understand the paper. However, I do believe that the paper would benefit from moderate English language editing prior to publication.